# How do patients make decisions in the context of a multidisciplinary team: an ethnographic study of four head and neck cancer centres in the north of England

David Winston Hamilton ,[1,2] Benjamin Heaven,[1] Richard Thomson,[1] Janet Wilson,[1] Catherine Exley[1]

¹Population Health Sciences Institute, Newcastle University, Newcastle upon Tyne, UK
²ENT, Newcastle upon Tyne Hospitals NHS Foundation Trust, Newcastle upon Tyne, UK

**Correspondence to**
David Winston Hamilton;
david.hamilton@ncl.ac.uk

## ABSTRACT

**Objectives** To describe how patients are engaged with cancer decisions in the context of multidisciplinary team (MDT) and how MDT recommendations are operationalised in the context of a shared decision.

**Design** Ethnographic qualitative study.

**Setting** Four head and neck cancer centres in the north of England.

**Participants** Patients with a diagnosis of new or recurrent head and neck cancer; non-participant observation of 35 MDT meetings and 37 MDT clinics, informal interviews, and formal, semistructured interviews with 20 patients and 9 MDT staff members.

**Methods** Ethnographic methods including non-participant observation of MDT meetings and clinic appointments, informal interviews, field notes and formal semistructured interviews with patients and MDT members.

**Results** MDT discussions often conclude with a firm recommendation for treatment. When delivered to a patient in clinic, this recommendation is often accepted by the patient, but this response may result from the disempowered position in which they find themselves. While patient behaviour may thus appear to endorse clinicians' views that a paternalistic approach is desired by patients (creating a 'cycle of paternalism'), the rigidity of the MDT treatment recommendation can act as a barrier to discussion of options and the exploration of patient values.

**Conclusions** The current model of MDT decision-making does not support shared decision-making and may actively undermine it. A model should be developed whereby the individual patient perspective has more input into MDT discussions, and where decisions are made on potential treatment options rather than providing a single recommendation for discussion with the patient. Deeper consideration should be given to how the MDT incorporates the patient perspective and/or delivers its discussion of options to the patient. In order to achieve these objectives, a new model of MDT working is required.

## INTRODUCTION

Multidisciplinary team (MDT) decision-making is internationally mandated to support appropriate high-quality treatment of patients with cancer.[1] In the UK, MDT working was established following the

---

### STRENGTHS AND LIMITATIONS OF THIS STUDY

⇒ This ethnographic study provides an in-depth analysis of the complexities of patient involvement and interaction with multidisciplinary team (MDT) decision-making.

⇒ The methods (direct observation and semistructured interviews) allow a rich, data-driven analysis of a complex decision-making environment.

⇒ Head and neck cancer involves the trade-off of function for survival and is thus a useful model when exploring complex decision-making.

⇒ As is commonplace in qualitative researcher, one researcher led the sampling, collecting and analysis, but the whole team were involved in discussions about interpretation of the data.

⇒ Although the structure of MDT decision-making discussed here predominates in the UK, the issues faced will not be applicable to all teams.

---

Calman-Hine report[2] and improves many aspects of cancer treatment such as staging, recruitment to trials,[1] adherence to treatment guidelines,[3] use of effective evidence-based therapy, timeliness of care[4] and access to the allied members of the healthcare team.[5] However, the practice is time-consuming and expensive, costing at least £100 million a year in the UK for data preparation and the same amount again for attendance in the UK.[6] To date, no MDT cost–benefit analysis has been performed.

MDT members report that consideration of the patient as a person in decision-making is a vital part of the decision-making process. In a survey of 2054 MDT members, 95% of respondents felt that 'Patient views should always inform the decision-making process' and 'Patient views/preferences should be presented to the MDT by somebody who has met the patient'.[7] Omitting patient preference information has an effect on the implementation of MDT recommendations.[8–10]

MDT meetings are often dominated by discussion among doctors rather than including other MDT members who may know the patient better or have a more patient-centred perspective,[11–14] creating a predominance of the biomedical model of disease.[15–17] This means that the stated aim of many MDT members—to have the patient central to the MDT treatment discussion—is at odds with the reality of the MDT process.

We have previously described that if MDT meetings are to become more patient centred, merely introducing increasing amounts of information about the patient into the MDT is not sufficient.[18] Although we know that the direct viewpoint of the patient within the MDT is lacking,[19] there is to date no account of how patients engage with decisions about their treatment in the context of MDTs. This work aims to explore the experience of making decisions in the context of an MDT, with a particular emphasis on the patient experience of the decision process.

## METHODS

This qualitative study used non-participant observation and semistructured interviews to critically examine how decisions were made in and around the MDT with a particular focus on patient centredness. All data were collected by one researcher (DWH), a head and neck surgeon. Non-participant observation enables the researcher to study participants in their natural environment, and adds value to retrospective accounts gleaned only through participant interviews.[20]

### Patient and public involvement

Two head and neck patient groups were consulted during development of the research question, study design and protocol development, but patients were not involved in data gathering and analysis.

### Sampling

Initial sampling aimed to recruit patients who had a treatment decision to make about their care, or where more than one treatment option was available to the patient. Concepts arising from the patient-derived data drove the subsequent data collection and analysis. A range of staff members who were part of the MDT were also recruited for interview. Purposive sampling[21] guided the sampling strategy to explore emerging concepts with data collection and analysis occurring in tandem. Cases were included which would test the concepts and themes which were emerging. For example, in the early cases, palliative options were often not discussed or offered in the clinic, and so patients who had options for treatment, one of which was palliative, were included. Concepts explored through sampling also included uncertainty, assessment of best and trust. Thus, further sampling was guided by the emerging analysis[22] and continued until a state of theoretical sufficiency[23] was achieved. This means that data collection ceases when sufficient or adequate depth of understanding has been reached; this allows for a greater number and breadth of concepts to be explored in this complex setting using multiple data collection techniques.

### Observations

Non-participant observations of 35 MDT meetings and 37 MDT outpatient clinics were conducted. Patients with a diagnosis of new or recurrent head and neck cancer whose treatment options were being discussed in the MDT were included. They were excluded if they did not understand written or spoken English, or they did not have the capacity to consent. The MDT meetings and clinics were all audio-recorded and transcribed verbatim. Detailed field notes were also made at the time of observation, then transcribed immediately afterwards.

### Interviews

Semistructured interviews were conducted with patients and staff. The development of the interview guide was iterative; as data collection continued, the content of the guide evolved in order to explore emerging themes.[24] In particular, the interview guide evolved to explore concepts of uncertainty (and how it is communicated), conversations around and attitudes towards palliative care, trust (between members of the MDT and between doctor and patient) and risk communication (see online supplemental file 1). Informal interviews with staff members of the MDT also took place and were incorporated into written field notes. Pseudonyms are used for reporting data throughout to protect the anonymity of respondents.

### Analysis

The data were analysed by one researcher (DWH) and emerging analyses were discussed with CE and BH, following principles of constructivist grounded theory.[21] Only one coder was used because of the complexity of the multiple data sources during this ethnographic study. However, emerging concepts and themes were discussed formally in the wider research team. All data sources (MDT meeting, clinic, informal and formal interviews) were analysed using the same coding framework. The codes used were conceptual, rather than descriptive, and labels were derived completely from the data, not predetermined. Line-by-line coding produced an initial coding framework: the emerging analysis was used during axial coding to guide further sampling and further development of the coding framework. Hence, coding was both inductive and deductive and when the coding framework was altered, all transcripts were recoded. The coding was organised using the NVivo computer package. Emerging findings (and 'memos') were formally discussed in the research team to develop the data analysis and guide subsequent analysis and data sampling.

## RESULTS

The research was conducted in three head and neck cancer centres in the northeast of England. A total of 35 MDT meetings and 37 clinic appointments (MDT

meetings and clinics) were observed for 30 patients (23 men and 7 women, aged 38–87 years). Additionally, 23 interviews were conducted with patients and 9 interviews with MDT members (see table 1). In all centres, the MDT meeting took place without the patient present and was attended by surgeons, oncologists, radiologists, pathologists, speech and language therapists, dietitians and administrative staff. Following the meeting, one surgeon met with the patient in clinic. Sometimes other members were present with the surgeon, and other times they were alone. If considering non-surgical options, the patient would meet an oncologist. Each MDT would discuss between 10 and 30 patients; the majority of these patients were then seen in the accompanying clinic.

## 'Best' treatment

The MDT meeting discussion often tends towards debate on which treatment is 'best' for a patient among the available options. In the following interview extract, a maxillofacial surgeon describes his view of the aim of the MDT discussion:

> [The team] need to leave the MDT [meeting] with the treatment options ….prioritised. So a rank order of [the] best treatment clinically – slightly irrespective of the patient's wishes. From a clinical point of view to try and get best outcome, this would be our first, this would be our second, this would be third and fourth and fifth. Then you discuss it with the patient and say, "This is what we think."

In this data extract, the surgeon clearly states his view that the aim of the MDT discussion is to decide the 'clinically' best treatment for the patient and even goes as far to say that this could be 'irrespective' of the patient's wishes. Teams frequently conclude their discussion of treatment options in the MDT meeting with an agreement for the recommendation (ie, the MDT's perception of 'best' treatment). This recommendation is to be delivered to the patient. In the following extract, the MDT members are discussing the merits of surgery (laser) versus radiotherapy.

> ENT surgeon 1: I have a database of the [laser resections] I have done …. tonsil and soft palate tumours, and it's just….it's something we need to take notice of.
> ENT surgeon 2: Yeah, I think we'll need to, we'll have to discuss it another time or we'll take up the whole morning on one case. But, I think there are arguments for and against…
> Oncologist 1: I would suggest he has radiotherapy, because he will have a slightly better functional outcome, and he's 80 and …because of his age, and because of the possibly better function….would you Dr Yellow?
> Oncologist 2: Yes.

> ENT surgeon 2: I think there is a consensus view of the MDT, would be for radiotherapy.
> ENT surgeon 1: OK.
> (Observation, MDT meeting)

Although during this discussion, options of radiotherapy and laser were available to the patient, the position of the MDT meeting was to provide a recommendation for radiotherapy. Here, we see the members of the MDT preparing their 'party line' which is to be delivered to the patient in clinic. This recommendation for 'best treatment' is often conveyed to the patient on its own or in preference to other options.

## The 'cycle of paternalism'

Anxious patients, faced with complex decisions, can lead to patients endorsing the paternalistic approach as they are given little or no information about the available treatment choices and therefore tend to delegate responsibility of the decision to the clinician:

> Patient: You know, they're the doctors, they're the professional people. And I'm just Joe Bloggs off the street. …For a lot of years, I was a steel erector. I wouldn't expect you as a doctor coming along and doing what I could do. Do I?
> Interviewer: You feel a decision should be the doctor's decision?
> Patient: Oh, definitely, without a doubt. It's got to be the doctor's decision. How could I make a decision like that?

Here, the patient delegates the decision to the MDT without question; he is allowing the MDT's assessment of 'best' treatment to act as the sole basis for a treatment decision. If decision delegation is accepted as the method by which MDTs convey and make decisions, a paternalistic decision-making process results. In this model, the patient accepts that the MDT's assessment of 'best' (and hence the treatment recommendation) is appropriate. It creates a 'cycle of paternalism' with grateful patients accepting firm recommendations from clinicians and clinicians reassured that they are doing their best for their patients.

## Delivery of the MDT treatment recommendation

Firm MDT recommendations can sometimes place the clinician in a difficult position when discussing options. The following extract is the clinic appointment for patient 6 (the MDT meeting extract was included above). Here, the ear, nose and throat (ENT) surgeon (who favoured laser in the MDT meeting, referred to here as 'surgery') was delivering the MDT recommendation for radiotherapy to the patient:

> ENT surgeon: After a lot of discussion, the consensus…. would be to give you radiation therapy…. that was what we jointly decided. And we think with that

**Table 1** Details of included participants

| Patients: group 1 | Centre | Age | Tumour site | Observation | | Interview 1 | Interview 2 |
| | | | | MDT | Clinic | | |
|---|---|---|---|---|---|---|---|
| Patient 1 | A | 68 | Pharynx | 1 | 1 | 1 | 1 |
| Patient 2 | A | 82 | Pharynx | 1 | 1 | x | x |
| Patient 3 | A | 61 | Parotid | 1 | 1 | x | x |
| Patient 4 | A | 71 | Lip | 1 | 1 | x | x |
| Patient 5 | A | 54 | Pharynx | 1 | 1 | 1 | x |
| Patient 6 | A | 80 | Pharynx | 1 | 1 | x | x |
| Patient 7 | A | 72 | Pinna | 1 | 1 | x | x |
| Patient 8 | A | 87 | Pharynx | 1 | 1 | 1 | x |
| Patient 9 | A | 64 | Larynx | 1 | 1 | 1 | x |
| Patient 10 | A | 61 | Larynx | 1 | 1 | 1 | x |
| Patient 11 | A | 52 | Pharynx | 1 | 1 | x | x |
| Patient 12 | A | 55 | Pharynx | 1 | 1 | 1 | 1 |
| Patient 13 | A | 62 | Larynx | 1 | 1 | 1 | x |
| Patient 14 | B | 73 | Pharynx | 3 | 1 | 1 | x |
| Patient 15 | B | 49 | Pharynx | 1 | 1 | 1 | x |
| Patient 16 | B | 52 | Pharynx | 1 | 1 | 1 | x |
| Patient 17 | B | 63 | Larynx | 1 | 1 | x | x |
| Patient 18 | B | 49 | Larynx | 1 | 1 | x | x |
| Patient 19 | B | 73 | Mouth | 3 | 1 | 1 | x |
| Patient 20 | B | 65 | Larynx | 1 | 1 | 1 | x |
| Patient 21 | B | 57 | Pharynx | 1 | 1 | x | x |
| Patient 22 | B | 63 | Pharynx | 1 | 2 | 1 | 1 |
| Patient 23 | C | 69 | Pharynx | 1 | 2 | 1 | x |
| Patient 24 | C | 81 | Mandible | 1 | 2 | x | x |
| Patient 25 | C | 60 | Pharynx | 1 | 1 | 1 | x |
| Patient 26 | C | 67 | Pharynx | 1 | 1 | x | x |
| Patient 27 | C | 46 | Pharynx | 1 | 2 | x | x |
| Patient 28 | C | 38 | Larynx | 1 | 1 | x | x |
| Patient 29 | C | 70 | Larynx | 1 | 4 | 1 | x |
| Patient 30 | C | 84 | Larynx | 1 | 1 | x | x |
| **Patients: group 2 (interview only)** | | | | | | | |
| Patient 31 | A | 82 | Pharynx | | | | |
| Patient 32 | A | 57 | Larynx | | | | |
| Patient 33 | A | 52 | Pharynx | | | | |
| Patient 34 | B | 65 | Larynx | | | | |
| **Staff (interview only)** | | **Staff role** | | | | | |
| Staff 1 | A | ENT surgeon | | | | | |
| Staff 2 | A | Oncologist | | | | | |
| Staff 3 | A | Maxillofacial surgeon | | | | | |
| Staff 4 | A | Speech and language therapist | | | | | |
| Staff 5 | A | Clinical nurse specialist | | | | | |
| Staff 6 | B | Maxillofacial surgeon | | | | | |
| Staff 7 | B | ENT surgeon | | | | | |
| Staff 8 | B | ENT surgeon | | | | | |
| Staff 9 | C | Oncologist | | | | | |

ENT, ear, nose and throat; MDT, multidisciplinary team.

treatment there is a very good chance of controlling your disease completely….

Patient: Well, I'll do as you say.

Daughter: So there's no other operation, it would just be radiotherapy?

ENT surgeon: We discussed this at length at the meeting…. and the majority of people… felt that to be frank, except for me, felt that radiation would be the way forward. And…. that's what we are offering to you as first line treatment. Unless you have any reservations, then we can think about other options.

Patient: I'll do as you say….

Daughter: Right. So he would have to come into hospital every day? He's a really bad traveller…

Patient: You know when I come here I get all tensed up and travelling….

ENT surgeon: Really? Is it likely you may then stop the treatment midway for whatever reason, because that would backfire very badly.

Patient: I wouldn't do that.

ENT surgeon: I know you asked about the surgical option. I promised people I wouldn't say anything. But it is feasible to take it out surgically, and there is an option available, but the consensus at the MDT was to go ahead with radiation. Unless, as a family or yourself very strongly object to it and feel that you can't go ahead with that, then of course the surgical option is always there. But as a group we felt that the best way forward was to offer you radiation.

Patient: Well. I'll go with you.

(Observation, MDT Clinic)

The final treatment decision was to deliver radiation, but the interaction above reveals the challenges of being tasked by the MDT to give a single recommendation when it is used in a decision discussion with a patient. Once new information was gleaned from the patient in clinic (being a 'really bad traveller'), the surgeon struggled with how to deal with the recommendations: was it a rule to be followed? Here, the rigidity of the treatment recommendation acted as a barrier to an open discussion about the treatment options available to the patient and thus inhibited shared decision-making.

### Patient engagement with MDT recommendations

Modern clinical practice cannot assume that the sole role of the patient is the acceptance of a single firm treatment recommendations. Patient 10 was a 61-year-old patient with an advanced cancer of his larynx. In the MDT, it was decided that surgery (total laryngectomy) should be delivered as a single recommendation. Radiotherapy is available, carries a lower chance of cure, but allows him to retain his voice box. The following data are from his clinic appointment:

ENT surgeon: This tumour in your throat is a fairly big tumour, and it's spread to the neck as well. We believe that there are two possible ways that we can manage this. At some parts of the scan, there is evidence that the tumour may have gone into the Adam's apple cartilage…. If that is the case, surgery would be the only option to get rid of the tumour. But surgery would involve you losing your voicebox, losing part of the swallowing passage, you would need a big neck operation….Once we do the surgery, your speech will be different, you won't be speaking the same. You will have a hole in the centre of your neck, a tracheostomy.

Patient: Nah, nah [shakes head].

ENT surgeon: You wouldn't fancy that?

Patient: No.

ENT surgeon: That's the surgical option. On the other side is the option of radiation therapy.

Patient: I would rather take a chance with that.

(Observation, MDT Clinic)

Here, and throughout the course of this consultation, the patient made a decision to reject surgery, which reduces his length of survival from his cancer in order to preserve his voice box. He was adamant he did not want a complete removal of the voice box and part of the throat (pharygolaryngectomy) and the decision was eventually made to use radiotherapy. However, in the subsequent interview, patient 10 discussed the rationale for his decision:

Patient: Well you see my mother died of cancer… my father died of cancer, and I've seen the way cancer works. I'm not being cheeky…. once they cut you open, it's like your letting fresh air into a bulb, it then just spreads, and they stitch you back up again and "We've cured it", right, for how long? And then it comes back again…

Interviewer: And what's important to you when you're making that decision?

Patient: Surviving as long as I can…, I mean if you get the year, 18 months it's better than getting two weeks isn't it?

His consultation, which was limited in exploring what mattered to him, leads to a decision that is potentially at odds with his aspirations revealed above. His aim of treatment (survival) is not matched by the actual treatment decision (radiotherapy). This patient perspective could not be incorporated into the previous MDT discussion (which happened before the clinic appointment), but equally the subsequent clinic appointment did not explore his preferences and what underpinned them, risking a treatment decision at odds with his preferences and values. If the surgeon had explored the options for treatment with the patient more, this mismatch of treatment preferences and values could have been identified,

and perhaps deconstructed. Such information about values and preferences is essential to good shared decision-making however very difficult to incorporate into the MDT decision-making structure.

## DISCUSSION

This study has found that patient engagement with the outcome of an MDT discussion (a recommendation for 'best' treatment) is problematic. Often patients accept this recommendation in the clinic (perhaps precisely because it is presented as the 'best' treatment). However, this acquiescence may be due to the disempowered position in which patients find themselves as they confront a terrifying diagnosis and a myriad of complex decision options. In turn, clinicians often view the acceptance of an MDT recommendation as delegation of the decision by the patient to the clinician, an assumption which can promulgate a 'cycle of paternalism', where anxious patients have little real choice other than to accept the clear guidance offered by the expert team. However, limiting patient involvement to acceptance or rejection of a firm recommendation leads to decisions which are not in line with patient values and cannot be considered patient-centred, shared decision-making.

The rigidity of the MDT recommendation can act as a barrier to an open discussion of the available options. If the patient role is limited to either acceptance or refusal of a single recommendation, true engagement is impossible. A truncated discussion of a single MDT recommendation for treatment prohibits shared decision-making using the 'three talk model', as central to this model is a discussion of the options for treatment. A shared decision-making consultation allows the patient and clinician to explore the risks, benefits and consequences of a treatment alternatives; a move from initial to informed preferences; and exploration of patient values to reach a shared decision.[25]

The structure of MDT working has not significantly changed since its inception in 1996. National Health Service patients rarely attend their MDT meetings, modern cancer care mandates that all patients are discussed in this setting[26] and interventions to increase the number of patients discussed in an MDT are still sought after.[27]

### The MDT recommendation

If the MDT meeting and clinic follow a paternalistic pathway, the way in which their recommendation is used is clear: it is delivered to the patient with an assumption that it will be accepted. In the paternalistic tradition, physicians are considered to be best placed to evaluate the trade-offs and pitfalls of treatment, and applied these to the decision process based on their evaluation of the best interests of the patient.[28] However, often in cancer care (particularly head and neck cancer), treatment options are available for a patient: which of these is 'best' depends on the value you apply to the various

aspects of the treatment. For example, is the priority of treatment cure or preservation of quality of life? What functional impact will a patient endure to achieve tumour control? What aspects of functional decline (such as speech, swallow or aesthetics) are most important? The answers to these questions are based on values: clinicians and patients do not share values.[29–31] Thus, MDTs must ensure that treatment decisions are driven by patient values. Although patients may justifiably actively delegate some or all of the responsibility for the decision to the MDT members, at the same time, the MDT has a duty to ensure that this is not due to disempowerment or lack of access to the information required to take an active part in decision-making. Hence, the clinician has a role to, at the very least, support the patient to understand what is important to them before accepting the role as decision-maker on the patient's behalf.

Out with the MDT decision process, a treatment recommendation from an individual clinician can be modified depending on the ongoing interaction with the patient and the preferences expressed. An MDT recommendation, on the other hand, is problematic for MDT members who attempt to combine it with the values or preferences of the patient. Is it set in stone, an obligatory 'best' which must be adhered to? If the patient disagrees with the recommendation, what action should the MDT member take? In this way, MDT recommendations are inflexible, especially in the light of new information from the patient which was not clear or known in the MDT meeting. In other words, information about values and preferences is vital to a shared decision but difficult to incorporate into the MDT decision-making structure. As we have previously described,[18] MDTs often build the 'evidential patient' in the MDT meeting discussion. This may include information about a patient's values and preferences, but these are impossible to incorporate into a meeting discussion without the patient present and without making assumptions about the patient.

### Modernising MDT decision-making

If we are to modernise the MDT decision-making structure to improve patient involvement, the role of the MDT discussion and the structure of the clinic must recognise that patients often 'distribute' decisions. Rapley[32] describes how patients demonstrate a 'relational autonomy' by distributing their decision among people, encounters, places and information sources. Promoting relational autonomy means that involving patients in decisions requires more than presenting options and awaiting a verdict, instead emphasising the importance of the interaction with the clinician, encouraging questions, correcting misunderstanding, constructing preferences and allowing disagreement.[33] Indeed, the MDT decision-making structure gives ample opportunity for MDT members to distribute their decision among colleagues, but does not afford the same opportunity to patients.

If the patient is to be a true participant in shared decision-making, an alternative model of MDT decision-making is

required. Some teams have explored the idea of a patient attending their own MDT meeting, with many patients reporting a positive experience[34]: this idea is popular among patient advocates,[35] but clinicians have mixed views.[7 35 36] Small studies have concluded that patients attending their own MDT allows for better information giving[37 38] and the opportunity to ask questions and contribute information such as preference[39]; however, included patients may have higher health literacy[40] raising the possibility that including patients has potential to widen health inequality. MDT members often feel that patients attending their own meeting would inhibit the discussion and cause patient anxiety[35]; relationships within the MDT are often longstanding with pre-existing hierarchies which can present barriers to new user integration.[41] Nevertheless, if patients are to be included in MDT meetings, clarity is required on how patients, their supporters and healthcare teams are supported to make it a positive and worthwhile experience.[42]

Of key importance is that the MDT meeting is not a discussion of which option is 'best' for a particular patient, but should instead aim to determine which valid treatment options are available. In particular, palliative options (or options of 'doing nothing') are often inadequately explored.[43] Clinic structures should be flexible to allow patients to distribute their decision-making among information sources and people. The patient may be enabled to come to the initial consultation more informed and prepared for the discussion. There may be a role for pre-MDT clinic with the patient meeting a surgeon, oncologist or specialist nurse, or a post-MDT clinic to convey options and explore values and preferences, maybe with more than one clinician. The MDT meeting may take place in a small 'combined clinic' setting around the interaction with the patient. The MDT members provide support, resources and personnel to discuss the treatment options, communicate the risk and uncertainty, elicit values and explore them; a decision aid may support this work.[44] The team may consider providing an individual who is independent of the clinical team to act as a decision coach or navigator.[45] MDT members should be encouraged to update their training in supporting patients in shared decision-making, consent and communication. This study provides a novel and rich account of the difficulties that patients face when making a decision in the context of an MDT. Sampling included patients with a decision to make or options available, which potentially excluded more straightforward cases which may make up a lot of MDT workload. MDT decision-making is mandated internationally; however, the specific structure of the decision process varies widely. Although the structure presented here (MDT meeting without a patient present, recommendation delivered to the patient separately) is common, other models of MDT decision-making may not face similar challenges. Also, ethnographic methods, in providing depth to explore a smaller number of concepts in more detail, may lack the breadth of findings to make this piece of work widely applicable. Nevertheless, while the setting may not be universally generalisable, we hope that the emergent conclusions will be.

It is time for the development and design of alternative models of team decision-making which have a central role for the patient. Further work to develop new model of delivering team decision-making would be multifactorial, incorporating the development of the structure of the MDT meeting and clinic, support and training for MDT members and patients and the development of tools to be used in combination with team decisions. Qualitative approaches should explore stakeholders' views of intervention components, which should be co-designed with patients. Evaluation of such interventions requires novel trial design, comparing methods of decision-making and evaluating decision quality. MDT decision-making is now ubiquitous and therefore the urgent need of reform to meet the principles of shared decision-making should be a priority for clinical teams and cancer researchers.

**Contributors** DWH—protocol development, ethical approval, data gathering, data analysis and manuscript preparation. BH—protocol development, ethical approval, data analysis and manuscript preparation. RT—protocol development, data analysis and manuscript preparation. JW—protocol development, data analysis and manuscript preparation. CE—protocol development, ethical approval, data analysis and manuscript preparation. DWH is the guarantor and accepts full responsibility for the work and/or the conduct of the study, had access to the data, and controlled the decision to publish.

**Funding** This study represents independent research funded by the National Institute of Health Research (DRF-2010-03-54).

**Disclaimer** The views expressed are those of the author(s) and not necessarily those of the National Health Service, the National Institute of Health Research or the Department of Health.

**Competing interests** None declared.

**Patient and public involvement** Patients and/or the public were involved in the design, or conduct, or reporting, or dissemination plans of this research. Refer to the Methods section for further details.

**Patient consent for publication** Obtained.

**Ethics approval** All participants gave written informed consent to be interviewed. Observational data (MDT meeting and clinic) were audio-recorded, then the patient was approached, 2–3 days after the event, to consent for inclusion in the study. If they agreed to be included, then the recordings were transcribed word for word. If they refused, all data collected so far were securely destroyed. This consent procedure was developed to avoid approaching the patient on the day of the treatment decision when they were already being given a lot of information. Ethical approval was gained from the NHS Research Ethics Newcastle and North Tyneside 2 committee (reference 11/NE/0200) in September 2011 and all necessary local Research and Development governance permissions were obtained.

**Provenance and peer review** Not commissioned; externally peer reviewed.

**Data availability statement** No data are available. No additional data available.

**ORCID iD**
David Winston Hamilton http://orcid.org/0000-0002-9653-6453

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
