## [Reviewer comments · BMJ Open]

ARTICLE DETAILS

TITLE (PROVISIONAL)	How do patients make decisions in the context of a multidisciplinary team: an ethnographic study of four head and neck cancer centres in the north of England
AUTHORS	Hamilton, David; Heaven, Benjamin; Thomson, Richard; Wilson, Janet; Exley, Catherine

VERSION 1 – REVIEW

REVIEWER	Rocque, Gabrielle University of Alabama at Birmingham, Medicine, Division of Hematology and Oncology
REVIEW RETURNED	20-Mar-2022

GENERAL COMMENTS	This is an interesting article about the need to engage patients in multidisciplinary team decisions, which is a very important topic. The research leverages direct observations of multidisciplinary tumor boards, informal feedback, and formal interviews. While the topic is important, my enthusiasm is dampened by the lack of robust methods of evaluation. I have the following suggestions for the authors: 1. The abstract should include more clear description of the methodology. The authors note that there are interviews in the participant section, but this should be moved to a more thorough description of the methods for the study.2. The results in the abstract are very narrative, without clear referral to key themes or barriers that are typical of qualitative methodology. The reporting of the results could be more structured.3. In the introduction, it is unclear what the multidisciplinary decision process is. This could be the context of a multidisciplinary tumor board or a clinic that includes all the providers. The details of the specific tumor board with details of structure, participants, frequency, proportion of all patients presented, number of patients discussed per tumor board, and any other relevant details should be included.4. The authors provide a solid case for the need to understand how patients make decisions after multidisciplinary care in a tumor board.5. The use of non-participant observation is a strength. However, the qualitative methodology would be strengthened by use of multiple reviewer and other methods to triangulate observations and results. The authors should provide a code book and the interview guide as a supplemental.6. Please describe in more detail the purposive sampling. What was considered?7. What were the criteria used to identify theoretic sufficiency? This methodology should be more clearly described.
---

	8. There needs to be more details on how the guide evolved and what components were added. 9. The use of a single coder is problematic. There should be at least 2 coders with another person who resolved discrepancies for robust qualitative analysis. 10. There are many statements in the results section that lack supporting quotes. For example, the authors state that they make decisions “Irrespective” of patient wishes, but that is not reflected in the quote. 11. In the physician exchange, it is unclear what the Mr. Black and Mr. Red portion is including. This section does not really capture or display well what the authors stated above. 12. I have some concerns about the authors interpretation that the physicians have a paternalistic view when the example quote reflects a patient wanting the physician to make the decision. Typically, I think of the paternalistic view when the patient preference is not included. In this example, the patient has expressed a preference for physician-led decision making. This may occur for some patients and not others but isn’t necessarily paternalism. The authors frame this as negative, but it is consistent with patient’s wishes. 13. The capture of the patient, daughter, physician discussion is interesting, but the authors present a lot of speculation and interpretation in results. This might be better suited for the discussion. 14. In the example with Mr. Winton, I am not certain how this was reflective of the tumor board discussion. 15. Please provide references and more context for the paternalism discussion. The results with the quotes and stories shared in do not necessarily point to paternalism, rather trouble with communicating effectively. 16. There can be a lot of heterogeneity for tumor boards, and this is not discussed as a limitation.
--	---

REVIEWER	Heuser, Christian University Hospital Bonn, Center for Health Communication and Health Services Research
REVIEW RETURNED	13-Apr-2022

GENERAL COMMENTS	Dear authors, dear David Hamilton, thank you very much for the opportunity to read and comment on your interesting manuscript. This is a very relevant topic in oncology addressing the question of patient-centeredness in multidisciplinary team meetings/multidisciplinary tumor conferences. You have conducted very relevant research using appropriate qualitative methods. However, I have some minor concerns, which will be addressed below. Please allow me a general comment right at the beginning: normally I am not recommending my own research papers during a review process, but concerning this paper I have to do an exception. Some main literature is missing here, so I have recommended original articles from me and my colleges with focus on patient participation in multidisciplinary tumor conferences in Germany. I think that they might be interesting for you, as you discuss the presence of patients in MDTs in the discussion section. General
---

	1. I recommend updating your reference list (see comments no 2 and 3). Introduction: 2. Thank you for this well written introduction. Concerning references 14-17 there is more research from UK and Belgium concerning the role of psychooncology and nurses in MDTs. E.g.:  - Soukup, Tayana; Murtagh, Ged; Lamb, Benjamin W.; Green, James S. A.; Sevdalis, Nick (2021a): Degrees of Multidisciplinary Underpinning Care Planning for Patients with Cancer in Weekly Multidisciplinary Team Meetings: Conversation Analysis. In: Journal of multidisciplinary healthcare 14, S. 411–424. DOI: 10.2147/JMDH.S270394. - Horlait, Melissa; Baes, Saskia; Dhaene, Sophie; van Belle, Simon; Leys, Mark (2019): How multidisciplinary are multidisciplinary team meetings in cancer care? An observational study in oncology departments in Flanders, Belgium. In: Journal of multidisciplinary healthcare 12, S. 159–167. DOI: 10.2147/JMDH.S196660. - Horlait, Melissa; Regge, Melissa de; Baes, Saskia; Eeckloo, Kristof; Leys, Mark (2022): Exploring non-physician care professionals' roles in cancer multidisciplinary team meetings: A qualitative study. In: PloS one 17 (2), e0263611. DOI: 10.1371/journal.pone.0263611. 3. My main comment deals with the following sentence “Although we know that the direct viewpoint of the patient within the MDT is lacking, there is to date no account of how patients engage with decisions about their treatment in the context of MDTs.” I would recommend to include the discussion on patient participation in MDTs in the Introduction section. Research on this topic started in Australia (references 29 and 31) and since 2014 also in Germany. References and a discussion regarding this topic is missing in my opinion. I have listed some topics we did/do research on in Germany:  - General study introduction: https://pubmed.ncbi.nlm.nih.gov/30962228/ - General practice of patient participation in MDTs in Germany: https://pubmed.ncbi.nlm.nih.gov/34402196/ - Providers views on patient participation in MDTs (feasibility) and shared decision making: https://pubmed.ncbi.nlm.nih.gov/33273821/ ; https://pubmed.ncbi.nlm.nih.gov/33051128/ ; https://pubmed.ncbi.nlm.nih.gov/32419276/ - Associated factors of patient participation, patient outcomes: https://pubmed.ncbi.nlm.nih.gov/30961598/ ; https://pubmed.ncbi.nlm.nih.gov/34004041/ ; - Communication in MDTs between patients and providers: https://pubmed.ncbi.nlm.nih.gov/34953619/ ; - https://pubmed.ncbi.nlm.nih.gov/34103928/ ; https://pubmed.ncbi.nlm.nih.gov/30669033/ ; 4. Please describe the aim of your study more precisely. Methods: 5. Ethical approval: Did the participants give written informed consent? 6. Analysis: “The codes used were conceptual, rather than descriptive, and labels were derived completely from the data, not pre-determined.” Maybe you can think of using the phrase “deductive/inductive codes” to make this relevant aspect more clearly?
--	---

	Results: 7. Please provide a sample description of the discussed patients (demographics, clinical data), your observations (duration of the case discussions, how many cases per MDT) and your interview partners (demographics, discipline, duration of interviews, ...). You can also think of providing more information on your codes/categories, coding tree, number of derived codes, etc? 8. "In all centres, the MDT meeting took place without the patient present; following this, one or more members of the MDT met with the patient in clinic." Do patients rarely participate in MDTs in your region/in your hospital/in head and neck MDTs or other entities? 9. MDT recommendation for "best treatment": I would guess that the recommendations based on clinical oncological treatment guidelines? I think this would be an important information for the reader and you could give a reference to the guidelines? 10. "(we have described this further in our previous paper 18)." This belongs into the discussion section. 11. First sentence in the paragraph The "cycle of paternalism": I think this belongs more in the introduction and should be underlined with a reference. Maybe you can just delete the sentence? Discussion: 12. Please start the discussion section with a summary of your aim and the results of your study. After that you can give information and discuss your results. Thank you. 13. "NHS patients do not routinely attend their MDT meetings, modern cancer care mandates that all patients are discussed in this setting²⁵ and interventions to increase the number of patients discussed in an MDT are still sought after²⁶." Please give information on patient participation in MDTs. 14. "Modernising MDT decision-making": Please see my comment no. 3. Here you mentioned a lot of aspects, which we started to study on in our PINTU study from 2017-2021 in Germany (smaller MDT settings if patient do participate, patient outcomes e.g. fear or information needs, decision-making processes and communication during MDTs with patient participation). I would recommend to include the discussion on patient participation in MDTs in the Introduction and Discussion section. References and a discussion regarding this topic is missing in my opinion. I have listed some topics we did/do research on in Germany in comment no 3. Thank you. Sincerely yours, Christian Heuser
--	--

VERSION 1 – AUTHOR RESPONSE

Reviewer 1

1. The abstract should include more clear description of the methodology. The authors note that there are interviews in the participant section, but this should be moved to a more thorough description of the methods for the study.

A "Methods" section has been added to the abstract with the following text

“Ethnographic methods including non-participant observation of MDT meetings and clinic appointments, with informal interviews and field notes; and formal semi-structured interviews with patients and MDT members”

2. The results in the abstract are very narrative, without clear referral to key themes or barriers that are typical of qualitative methodology. The reporting of the results could be more structured.

Thank you for highlighting this. Some of the findings from this work have previously been reported. This paper focussed on the concept of “best”, how it drives the MDT recommendation. We then go on to discuss how the recommendation is delivered and how this impacts shared decision making. Some of the wording of the results section and headers have been changed to improve clarity, but the narrative of the results section has remained in order to maintain a strong message throughout the paper

3. In the introduction, it is unclear what the multidisciplinary decision process is. This could be the context of a multidisciplinary tumor board or a clinic that includes all the providers. The details of the specific tumor board with details of structure, participants, frequency, proportion of all patients presented, number of patients discussed per tumor board, and any other relevant details should be included.

As MDT and tumor boards vary throughout the world, the introduction has not mentioned a specific set up. But at the start of the results section, the opening paragraph has been edited as follows: “The research was conducted in three head and neck cancer (HNC) centres in the north east of England. In all centres, the MDT meeting took place without the patient present and was attended by surgeons, oncologists, radiologists, pathologists, speech and language therapists, dieticians and administrative staff. Following the meeting, one surgeon met with the patient in clinic. Sometimes other members were present with the surgeon, and other times they were alone. If considering non-surgical options, the patient would meet an oncologist. Each MDT would discuss between 10 and 30 patients; the majority of these patients were then seen in the accompanying clinic”

5. The use of non-participant observation is a strength. However, the qualitative methodology would be strengthened by use of multiple reviewer and other methods to triangulate observations and results. The authors should provide a code book and the interview guide as a supplemental.

The interview guide has been added as a supplemental

6. Please describe in more detail the purposive sampling. What was considered?

The following statement has been added to the sampling paragraph

“Cases were included which would test the concepts and themes which were emerging. For example, in the early cases, palliative options were often not discussed or offered in the clinic, and so patients were included who had options for treatment, one of which was palliative, were included. Concepts explored through sampling also included uncertainty, assessment of best and trust.”

7. What were the criteria used to identify theoretic sufficiency? This methodology should be more clearly described.

The following text has been added

“This means that data collection ceases when sufficient or adequate depth of understanding has been reached” We used this as it this allows for a greater number and breadth of concepts to be explored. The setting was complex and we were using multiple data collection techniques

8. There needs to be more details on how the guide evolved and what components were added.

The following statement has been added to the interviews paragraph

“In particular, the interview guide evolved to explore concepts of uncertainty (and how it is communicated), conversations around and attitudes towards palliative care, trust (between members of the MDT and between doctor and patient) and risk communication. “

9. The use of a single coder is problematic. There should be at least 2 coders with another person who resolved discrepancies for robust qualitative analysis.

As this was an ethnographic study, a single researcher was immersed in the setting, conducting and making field notes informally. Much of the transcription of the complex MDT meetings (with as many as 20-30 attendees) were performed by DWH. This gave an excellent knowledge of the data from multiple sources during this ethnographic study. In order to improve analysis, emerging concepts were discussed with CE and BH and memos were created and discussed. The following statement has been added to the analysis paragraph

“Only one coder was used because of the complexity of the multiple data sources during this ethnographic study. However emerging concepts and themes were discussed formally in the wider research team.”

10. There are many statements in the results section that lack supporting quotes. For example, the authors state that they make decisions “Irrespective” of patient wishes, but that is not reflected in the quote.

We have, over multiple revisions, attempted to get through a number of concepts (with data) within the word count; we accept that at times the data presented may not represent every aspect of the concept being explored, but that it is indicative. However, in the example above, Mr Halifax does in fact use the word “irrespective”. The whole quote is copied here (irrespective highlighted in red) [The team] need to leave the MDT [meeting] with the treatment optionsprioritised. So a rank order of [the] best treatment clinically – slightly irrespective of the patient’s wishes. From a clinical point of view to try and get best outcome, this would be our first, this would be our second, this would be third and fourth and fifth. Then you discuss it with the patient and say, “This is what we think.”

11. In the physician exchange, it is unclear what the Mr. Black and Mr. Red portion is including. This section does not really capture or display well what the authors stated above.

This extract is included to demonstrate how the members of the MDT prepare their ‘party line’ prior to seeing the patient, thus concealing any uncertainty (and hence choice) from the patient. We have explored the uncertainty and its relationship to choice more in our other paper on this subject.

<https://bmjopen.bmj.com/content/bmjopen/6/7/e012559.full.pdf>

We have added sentences to the preceding paragraph and the analysis paragraph to improve clarity

12. I have some concerns about the authors interpretation that the physicians have a paternalistic view when the example quote reflects a patient wanting the physician to make the decision. Typically, I think of the paternalistic view when the patient preference is not included. In this example, the patient has expressed a preference for physician-led decision making. This may occur for some patients and not others but isn’t necessarily paternalism. The authors frame this as negative, but it is consistent with patient’s wishes.

The following paragraph has been added to the discussion at the outset of the discussion about the MDT recommendation

“If the MDT meeting and clinic follow a paternalistic pathway, the way in which their recommendation is used is clear: it is delivered to the patient with an assumption that it will be accepted. In the paternalistic tradition, physicians are considered to be best placed to evaluate the trade-offs and pitfalls of treatment, and applied these to the decision process based on their evaluation of the best interests of the patient³⁰. However, often in cancer care (particularly head and neck cancer), treatment options are available for a patient: which of these is “best” depends on the value you apply to the various aspects of the treatment. For example, is the priority of treatment cure or preservation of quality of life? What functional impact will a patient endure to achieve tumour control? What aspects of functional decline (such as speech, swallow or aesthetics) are most important? The answers to these questions are based on values: clinicians and patient do not share values³¹⁻³³, thus MDTs must ensure that treatment decisions are driven by patient values. Although patients may justifiably actively delegate some or all of the responsibility for the decision to the MDT members, at the same time, the MDT have a duty to ensure that this is not due to disempowerment or lack of

access to the information required to take an active part in decision making. Hence the clinician has a role to, at the very least, support the patient to understand what is important to them before accepting the role as decision maker on the patient's behalf"

13. The capture of the patient, daughter, physician discussion is interesting, but the authors present a lot of speculation and interpretation in results. This might be better suited for the discussion. A section of the analysis of this quote has been removed and added to the discussion of paternalism in the discussion section

14. In the example with Mr. Winton, I am not certain how this was reflective of the tumor board discussion. The MDT recommendation from the meeting has been made clear in the preceding paragraph to put the comments from Mr Black in context

15. Please provide references and more context for the paternalism discussion. The results with the quotes and stories shared in do not necessarily point to paternalism, rather trouble with communicating effectively. The following paragraph has been added to the discussion of the MDT recommendation "Often in cancer care (particularly head and neck cancer), treatment options are available for a patient: which of these is "best" depends on the value you apply to the various aspects of the treatment. For example, is the priority of treatment cure or preservation of quality of life? What functional impact will a patient endure to achieve tumour control? What aspects of functional decline (such as speech, swallow or aesthetics) are most important? The answers to these questions are based on values: clinicians and patient do not share values³⁰⁻³². Thus MDTs must ensure that treatment decisions are driven by patient values. Although patients may justifiably actively delegate some or all of the responsibility for the decision to the MDT members, at the same time, the MDT have a duty to ensure that this is not due to disempowerment or lack of access to the information required to take an active part in decision making. Hence the clinician has a role to, at the very least, support the patient to understand what is important to them before accepting the role as decision maker on the patient's behalf"

16. There can be a lot of heterogeneity for tumor boards, and this is not discussed as a limitation. The last bullet in the "Strengths and limitations" section has been changed to "Internationally, there are multiple models of MDT decision making. Although the structure discussed here predominates in the UK, the issues faced will not be applicable to all teams" Also, a limitations paragraph has been added to the discussion section

"This study provides a novel and rich account of the difficulties that patients face when making a decision in the context of an MDT. MDT decision making is mandated internationally however the specific structure of the decision process varies widely. Although the structure presented here (MDT meeting without a patient present, recommendation delivered to the patient separately) is common, other models of MDT decision making may not face similar challenges. Also, ethnographic methods, in providing depth to explore a smaller number of concepts in more detail, may lack the breadth of findings to make this piece of work widely applicable. Nevertheless, whilst the setting may not be universally generalisable, we hope that the emergent conclusions will be."

Reviewer 2

Mr. Christian Heuser, University Hospital Bonn Comments to the Author:
Dear authors, dear David Hamilton,

General

1. I recommend updating your reference list (see comments no 2 and 3).

The reference list has been updated

Introduction:

2. Thank you for this well written introduction. Concerning references 14-17 there is more research from UK and Belgium concerning the role of psychooncology and nurses in MDTs.

Thank you for providing these references, we read them with interest and they have been used to reference the existing statements

3. My main comment deals with the following sentence “Although we know that the direct viewpoint of the patient within the MDT is lacking, there is to date no account of how patients engage with decisions about their treatment in the context of MDTs.” I would recommend to include the discussion on patient participation in MDTs in the Introduction section.

Thank you for the included studies, which we read with interest. As a result of this, the paragraph on patient participation in MDTs has been updated and the references have also been updated

“If the patient is to be a true participant in shared decision making, an alternative model of MDT decision making is required. Some teams have explored the idea of a patient attending their own MDT meeting, with many patients reporting a positive experience³²: this idea is popular amongst patient advocates³³, but clinicians have mixed views^{10 33 34}. Small studies have concluded that patients attending their own MDT allows for better information giving^{35 36} and the opportunity to ask questions and contribute information such as preference³⁷; however included patients may have higher health literacy³⁸ raising the possibility that including patients has potential to widen health inequality. MDT members often feel that patients attending their own meeting would inhibit the discussion and cause patient anxiety³³; relationships within the MDT are often longstanding with pre existing hierarchies which can present barriers to new user integration³⁹ Nevertheless, if patients are to be included in MDT meetings, clarity is required on how patients, their supporters and healthcare teams are supported to make it a positive and worthwhile experience⁴⁰”

4. Please describe the aim of your study more precisely.

The aim of the study has been updated to “This work aims to explore the experience of making decisions in the context of an MDT, with a particular emphasis on the patient experience of the decision process”

Methods:

5. Ethical approval: Did the participants give written informed consent?

All participants gave informed consent, however this was not written for all participants; for example, MDT participants changed week to week. All of them were informed about the study but not all MDT participants signed a written consent form. Our pragmatic consent processes were discussed and approved by an ethics committee

6. Analysis: “The codes used were conceptual, rather than descriptive, and labels were derived completely from the data, not pre-determined.” Maybe you can think of using the phrase “deductive/inductive codes” to make this relevant aspect more clearly?

The phrasing inductive and deductive coding have been added to the analysis paragraph to improve clarity

Results:

7. Please provide a sample description of the discussed patients (demographics, clinical data), your observations (duration of the case discussions, how many cases per MDT) and your interview partners (demographics, discipline, duration of interviews, ...).

The following text, and a table of included participants have been included:

“The research was conducted in three head and neck cancer (HNC) centres in the north east of England. A total of 35 MDT meetings and 37 clinic appointments MDT meetings and clinics were observed for 30 patients (23 males and seven females, aged 38-87 years). Additionally 23 interviews were conducted with patients and nine interviews with MDT members. “

8. “In all centres, the MDT meeting took place without the patient present; following this, one or more members of the MDT met with the patient in clinic.” Do patients rarely participate in MDTs in your region/in your hospital/in head and neck MDTs or other entities?
In the UK, it is extremely rare for the patient to attend the MDT meeting

9. MDT recommendation for “best treatment”: I would guess that the recommendations based on clinical oncological treatment guidelines? I think this would be an important information for the reader and you could give a reference to the guidelines?

Although guidelines exist for head and neck cancer management, there exists considerable uncertainty about which treatment is “best”. This means that many of the guidelines have more than one treatment option available for a patient with a particular stage of disease To demonstrate this, the following paragraph has been added

“Often in cancer care (particularly head and neck cancer), treatment options are available for a patient: which of these is “best” depends on the value you apply to the various aspects of the treatment. For example, is the priority of treatment cure or preservation of quality of life? What functional impact will a patient endure to achieve tumour control? What aspects of functional decline (such as speech, swallow or aesthetics) are most important? The answers to these questions are based on values: clinicians and patient do not share values ³⁰⁻³². Thus MDTs must ensure that treatment decisions are driven by patient values. Although patients may justifiably actively delegate some or all of the responsibility for the decision to the MDT members, at the same time, the MDT have a duty to ensure that this is not due to disempowerment or lack of access to the information required to take an active part in decision making.”

10. “(we have described this further in our previous paper 18).” This belongs into the discussion section.

This sentence has been deleted from the results section

11. First sentence in the paragraph The “cycle of paternalism”: I think this belongs more in the introduction and should be underlined with a reference. Maybe you can just delete the sentence?
The sentence has been deleted and the start of the paragraph altered

Discussion:

12. Please start the discussion section with a summary of your aim and the results of your study. After that you can give information and discuss your results. Thank you.
The first paragraph of the discussion has been altered to give a summary of the findings

13. “NHS patients do not routinely attend their MDT meetings, modern cancer care mandates that all patients are discussed in this setting and interventions to increase the number of patients discussed in an MDT are still sought after.” Please give information on patient participation in MDTs. The sentence has been altered to say “NHS patients rarely attend their MDT meetings” (as this is the case) and the paragraph regarding patient participation in their own MDTs has been revised, as outlined above

14. “Modernising MDT decision-making”: Please see my comment no. 3. Here you mentioned a lot of aspects, which we started to study on in our PINTU study from 2017-2021 in Germany (smaller MDT settings if patient do participate, patient outcomes e.g. fear or information needs, decision-

making processes and communication during MDTs with patient participation). I would recommend to include the discussion on patient participation in MDTs in the Introduction and Discussion section. References and a discussion regarding this topic is missing in my opinion. I have listed some topics we did/do research on in Germany in comment no 3. Thank you.

The paragraph on patient participation in MDT decision making has been revised as above and is much improved as a result, thank you

I hope that the changes meet with the approval of you and your reviewers and I am very grateful for the review and comments

VERSION 2 – REVIEW

REVIEWER	Rocque, Gabrielle University of Alabama at Birmingham, Medicine, Division of Hematology and Oncology
REVIEW RETURNED	07-Jun-2022

GENERAL COMMENTS	Thank you for the response to reviews.  1. I am struck by the commentary about relationship between MTD survival in the introduction, which doesn't seem aligned with the goal of increasing patient centricity. I would remove this and save space. 2. The purposive sampling doesn't really make sense for the tumor boards. I do understand why you couldn't review all cases. I am concerned that this sampling method could introduce bias. 3. I continue to have concerns that the data was only reviewed by one coder. 4. Please remove all patient names. Even if pseudonyms in the table, this is confusing and should be changed to Patient 1, 2, ect.
---

REVIEWER	Heuser, Christian University Hospital Bonn, Center for Health Communication and Health Services Research
REVIEW RETURNED	31-May-2022

GENERAL COMMENTS	Dear Mr. Hamilton, thank you very much for your detailed response to my comments. All of my comments have been answered satisfactorily. Best regards, Christian Heuser
---

VERSION 2 – AUTHOR RESPONSE

Reviewer 1

1. I am struck by the commentary about relationship between MTD survival in the introduction, which doesn't seem aligned with the goal of increasing patient centricity. I would remove this and save space.

The sentence about survival has been removed "The effect on cancer survival is less clear; in head and neck cancer MDT working is reported to have a positive effect on survival ⁷⁻⁹, but it is difficult to determine whether survival changes over time are causally related."

2. The purposive sampling doesn't really make sense for the tumor boards. I do understand why you couldn't review all cases. I am concerned that this sampling method could introduce bias.

3. I continue to have concerns that the data was only reviewed by one coder.

To attempt to address both of these concerns (and respond to editorial comments), point four of the strengths and limitations section has been altered to "As is commonplace in qualitative researcher one researcher led the sampling, collecting and analysis, but the whole team were involved in discussions about interpretation of the data"

4. Please remove all patient names. Even if pseudonyms in the table, this is confusing and should be changed to Patient 1, 2, ect.

The patient names have been changed to patient numbers as described in table one and the text of the results section

I hope that the changes meet with the approval of you and your reviewers and I am very grateful for the review and comments

Yours sincerely

David Hamilton

Corresponding author